# Distribution of Iron, Copper, Zinc and Cadmium in Glia, Their Influence on Glial Cells and Relationship with Neurodegenerative Diseases

**DOI:** 10.3390/brainsci13060911

**Published:** 2023-06-05

**Authors:** Aleksandra Górska, Agnieszka Markiewicz-Gospodarek, Renata Markiewicz, Zuzanna Chilimoniuk, Bartosz Borowski, Mateusz Trubalski, Katarzyna Czarnek

**Affiliations:** 1Department of Human Anatomy, Medical University of Lublin, 4 Jaczewskiego St., 20-090 Lublin, Poland; agnieszkamarkiewiczgospodarek@umlub.pl; 2Department of Psychiatric Nursing, Medical University of Lublin, 18 Szkolna St., 20-124 Lublin, Poland; renatamarkiewicz@umlub.pl; 3Student Scientific Group at the Department of Family Medicine, 6a (SPSK1) Langiewicza St., 20-032 Lublin, Poland; zuzia.chil@gmail.com; 4Students Scientific Association at the Department of Human Anatomy, Medical University of Lublin, 20-090 Lublin, Poland; bartosz.borowski@interia.pl (B.B.); mateusztrub@gmail.com (M.T.); 5Institute of Health Sciences, The John Paul II Catholic University of Lublin, Konstantynów 1 H, 20-708 Lublin, Poland; katarzyna.czarnek@kul.pl

**Keywords:** Zn, Cu, Fe, Cd, neurodegenerative diseases, Alzheimer disease, multiple sclerosis, Parkinson disease, amyotrophic lateral sclerosis

## Abstract

Recent data on the distribution and influence of copper, zinc and cadmium in glial cells are summarized. This review also examines the relationship between those metals and their role in neurodegenerative diseases like Alzheimer disease, multiple sclerosis, Parkinson disease and Amyotrophic lateral sclerosis, which have become a great challenge for today’s physicians. The studies suggest that among glial cells, iron has the highest concentration in oligodendrocytes, copper in astrocytes and zinc in the glia of hippocampus and cortex. Previous studies have shown neurotoxic effects of copper, iron and manganese, while zinc can have a bidirectional effect, i.e., neurotoxic but also neuroprotective effects depending on the dose and disease state. Recent data point to the association of metals with neurodegeneration through their role in the modulation of protein aggregation. Metals can accumulate in the brain with aging and may be associated with age-related diseases.

## 1. Introduction

The human body consists of many chemical elements, which can be divided into trace- and macroelements. Iron, copper and zinc are some of the trace elements that are present in every human body, as they play a role in some important mechanisms [1,2,3,4,5,6,7]. Amongst the many functions, these trace elements are also significant in the neural system functioning. Iron is important in the neurotransmitter metabolism and takes part in myelin synthesis [8]. Copper is crucial for many enzymes to work properly, and some of these enzymes play important roles in the central nervous system. The abnormalities in these enzymes, caused by copper deficiency, may lead to some diseases [9,10]. Zinc is also a cofactor for some enzymes, for example DNA-polymerase, which are also needed for the nervous system to develop properly, as they are required for neurogenesis [11]. Low zinc levels can lead to such neurological abnormalities as cognitive disorders or mental health issues [12].

On the other hand, some trace elements are not normally present in the human body, and their presence may lead to some disfunctions. For example, cadmium is an element that should not be part of a healthy organism as it can cause some adverse effects [13]. This heavy metal is a common pollutant, and due to the possibility of it accumulating in brain tissue, it may cause cognitive function disorders and abnormalities in the physiological enzymes’ functioning [14].

It has been proven that the concentration of the elements varies in different brain regions. Oligodendrocytes, which are involved in myelin synthesis, were shown to have the highest concentration of iron among all the glial cell types [15,16]. There have been similar observations regarding other metals such as zinc and copper [17,18,19]. Glial cells, including oligodendrocytes and astrocytes, play a crucial role in the regulation, distribution and storage of metals such as iron, copper and zinc in the brain [19,20,21]. Imbalances in metal levels within glial cells may result in significant implications for brain function and contribute to neurodegenerative diseases [22,23]. In addition, many papers have stated that there is a connection between trace elements, composition dysfunctions and neurodegenerative diseases [24,25,26,27,28,29,30]. Research regarding whether the regulation of the metals’ level might help with these diseases’ treatment is ongoing [26,31,32,33]. 

Importantly, neurodegenerative diseases, such as Parkinson’s disease (PD), Alzheimer’s disease (AD), multiple sclerosis (SM) and amyotrophic lateral sclerosis (ALS), are a common problem in society. In the United States of America, there are an estimated 6.7 million people aged 65 or older who are affected by AD, and this number is still growing [34]. The total cost generated by AD itself is estimated to reach more than USD 1 trillion in 2050 [35]. These are some of the reasons for the growing interest in neurodegenerative diseases, their pathogenesis, diagnostics, and treatment. It is important to find new methods that could help in the neurodegenerative diseases’ treatment, which remains a serious challenge for the physicians.

The aim of this study was to describe the distribution and influence of iron, copper, zinc and cadmium on glial cells. Moreover, we wanted to focus on the influence of those metals on neurodegenerative diseases like Alzheimer disease, multiple sclerosis, Parkinson disease and Amyotrophic lateral sclerosis. We hope that this review might broaden current knowledge about the role of metals in the pathophysiology of the brain and may be helpful for future studies on neurodegenerative diseases.

## 2. Distribution and Influence of Iron, Copper, Zinc and Cadmium on Glial Cells

### 2.1. Iron

Iron is present in abundant amounts in the brain, fulfilling pleiotropic functions including neuronal myelination and synthesis of neurotransmitters, DNA, RNA and proteins [36,37,38]. Thereby, iron is essential as a cofactor of numerous enzymes, especially for adenosine triphosphate (ATP) production [17]. A balanced iron level ensures that iron is kept in adequate concentration to fulfill its crucial functions in the brain while its harmful effects remain under control. The brain constantly needs available amounts of iron in a regional, cellular and age-sensitive mode [37]. Failure to meet this demand for iron can result in persistent neurological dysfunction. On the other hand, increased iron levels and iron accumulations in specific brain regions and cells are hallmarks for numerous neurodegenerative diseases [36,37,38,39,40,41,42]. The prominent neurodegenerative diseases with an iron-associated pathology are Alzheimer’s and Parkinson’s diseases. Iron also plays a role in the etiology of Huntington’s disease, multiple sclerosis and progressive supranuclear palsy. There is even a group of diseases referred to as NBIA (neurodegeneration with brain iron accumulation) [43].

Iron metabolism varies across different cell types such as neurons, astrocytes, oligodendrocytes and microglia due to their distinct structural and metabolic characteristics. Consequently, the presence of iron in affected brain regions may not indicate a general increase but rather the accumulation and redistribution of iron specific to certain cell types. Reinert et al. (2019) [44] published findings that demonstrated oligodendrocytes possess the highest iron concentration among glial cells, aligning with previous studies [15,16]. It is believed that the synthesis of myelin, which is a primary function of oligodendrocytes [45], necessitates elevated iron levels to fulfill their enzymatic and metabolic requirements. Additionally, oligodendrocytes likely play a role in iron regulation as they are equipped with iron storage and transport proteins [46]. Oligodendrocytes play a neuroprotective and supportive role or, in case of dysfunction, might initiate or progress the degeneration process [47]. For instance, in mice lacking the iron-responsive element, a model of neurodegeneration resulting from abnormal iron regulation, oligodendrocytes exhibit a significant increase in ferritin levels, while neighboring degenerating neurons experience a substantial decrease [48]. Regarding astrocytes, Hoepken et al. (2004) [49] employed atomic absorption spectroscopy (AAS) to measure iron content in cultured rat astrocyte lysates. Normalized to protein content, they obtained a value of (9.3 ± 5) nmol iron/mg protein. This aligns well with the in situ conducted by Reiner et al. (2019) and the results of (5.9 ± 8) nmoliron/mg protein. Another study by Bishop et al. (2011) investigated iron content in astrocytes derived from newborn mice using a ferrozine-based colorimetric assay [50]. Using a ferrozine-based colorimetric assay, they calculated, based on protein content, and estimated cytosolic volume, an intracellular iron concentration of (1.2 ± 6) mM (mean ± SD; n ≥ 3 cultures). Within the margin of errors, the iron concentration of mice primary astrocytes published by Bishop and coworkers is the same as Reinert et al., measured for rat astrocytes in situ ((1.29 ± 14) mM). However, due to the lack of knowledge on the cytosolic volume of neurons and microglia, it was not possible to calculate the iron concentration for these cell types. Nevertheless, Bisop et al. (2011) discovered that microglia and astrocytes accumulate more iron than neurons, with microglia exhibiting the highest efficiency. These findings align with other studies indicating that microglia tend to contain more iron than astrocytes (with a slight statistical significance of *p* < 0.07) and that neurons have the lowest iron content, thus highlighting similar patterns. 

Reiner et al. (2019) [44] demonstrated that neurons in the neocortex, substantia nigra, subiculum and deep cerebellar nuclei exhibit similar iron concentrations. However, the iron content differs between glial cells and neurons with glial cells generally containing more iron. Moreover, the iron concentration within glial cells varies depending on the specific type of glial cells. These findings suggest that the iron levels are influenced more by the type of neural cell and the metabolic characteristics and functions of the cell rather than the brain region in which the cell is situated.

Neural cell types can individually regulate the distribution and storage of iron according to their needs and functions [51]. For example, oligodendrocytes secrete transferrin to other cells, microglia provide iron to oligodendrocytes to obviously ensure their demand of iron [52], and astrocytes are known to regulate the transport of iron to other cells [53]. Consequently, maintaining a proper balance of cellular iron requires a regulated process of iron uptake, storage, distribution, and release, both within individual cells and between different cells. The intricate and precise functioning of iron metabolism in the brain is crucial, given its specialized nature. However, it is also a delicate process. Additionally, as individuals age, iron undergoes redistribution among different molecular forms, such as ferritin, neuromelanin, transferrin, and hemosiderin, resulting in changes in the distribution of iron between neurons and glial cells [40].

Tarohda et al. (2004) [54] studied age-related changes in metal concentrations in several brain regions of Wistar rats. They report increasing iron concentrations during postnatal development from P1 to P42, reaching adult levels before P72. These findings suggest that iron-related specializations and regulatory circuits have essentially developed within the first two months of age. The augmented iron levels can be attributed to the constant delivery of iron into the brain and decreased release of iron into the blood via the BBB and BCSFB [55]. It is tempting to postulate that iron may be drained from the cerebrospinal fluid (CSF) via the lymphatic system [17]. 

The accumulation of iron is a characteristic feature of aging, and it is linked to numerous age-related diseases. Iron plays a catalytic role in the formation of harmful cellular waste molecules and hinders their elimination. Conversely, reducing iron levels in the bloodstream could potentially have rejuvenating effects. Iron is intricately intertwined with the aging process, and maintaining control over the body’s iron stores may emerge as a significant approach to prolonging human lifespan [56].

Under the assumption that glial cells exhibit similar functions and physiology across different brain regions, it can be inferred that glial cells, particularly oligodendrocytes, are the brain’s most iron-rich cells. Therefore, when conducting studies that measure iron content in the brain at a resolution lower than the cellular level, it is crucial not only to focus on neurons but also to consider the contribution of glial cells to the overall iron levels. Moreover, imbalances in iron homeostasis that lead to neurodegeneration may not solely be localized in neurons but could also, or primarily, occur in glial cells. Recognizing the significance of glial cells in iron-related processes is essential for a comprehensive understanding of iron dynamics in the brain.

### 2.2. Copper

The brain contains the highest copper (Cu) content in the body after the liver [57]. Cu is required for modulation of synaptic activity and neuronal plasticity [58] Cu is a cofactor of proteins involved in neurotransmitter biosynthesis, mitochondrial activity, oxidative stress defense and other fundamental brain processes. Cu is crucial for physiological brain function [59]. On the other hand, an abnormal Cu level has been implicated in the pathogenesis of various neurodegenerative disorders, including Alzheimer’s disease, Menkes disease, Parkinson’s disease, familial amyotrophic lateral sclerosis, and prion disorders [22,23]. Despite its fundamental role in brain physiology, knowledge of Cu transport into the brain, its distribution and regulation are still limited. It was proved that Cu content is increased with aging [60]; however, no experimental procedure has been performed to show whether the observed increase is uniform among different brain areas and cells or whether some brain areas accumulate Cu to a much greater level. It has been hypothesized that among the parenchymal cells, astrocytes have the greatest influence on Cu homeostasis in the brain [19]. It was demonstrated that rat astrocytes in cultures efficiently take up Cu from medium [61]. Cell culture studies also indicate that neurons are more susceptible to Cu-induced cytotoxicity than astrocytes [62]. These studies were carried out in cell cultures because techniques for tracking Cu ions in particular brain cells were not accessible.

For the first time, implementation of X-ray fluorescence microscopy allowed analyzing the Cu distribution in rodent brains, to visualize intracellular Cu accumulations in glial cells, and to follow development of these accumulations with aging [20]. It was proved that live astrocytes have mechanism to detoxify accumulated Cu; however, if Cu-loaded astrocytes would die because of brain injury or aging, toxic Cu might be released into the environment and affect brain cells.

Ashraf et al. (2019) [17] proved that copper was remarkably enriched in the ventricles with aging, consistent with previous observations of increased copper at the choroid plexus [43]. It has been suggested that the blood cerebrospinal fluid barrier is the predominant barrier for regulated copper uptake in the brain [63]. Astrocytes in the locality of ventricles are in a prime location to balance brain copper content and carry out the detoxification, having access to cerebrospinal fluid (CSF) [20]. Like any mammalian cells, astrocytes uptake Cu into the cytoplasm by the copper transporter 1 (Ctr1). Ctr1 passes Cu to Cu chaperones, which mediate the intracellular transport of Cu to Cu proteins via mechanisms utilizing direct protein–protein interaction blood. Astrocytes have perivascular end feet or footplates, expanses of their cytoplasmic processes that surround the abluminal surfaces of the capillary endothelial cells that form the blood–brain barrier (BBB) of the brain and are opposed to the endothelial basal lamina [64]. The BBB is partially lost on the surface of blood capillaries in the subventricular zone (SVZ) [65], which can explain the highest Cu accumulations detected in astrocytes positioned close to the ventricle wall. The choroid plexus, a tissue in brain ventricles adjunct to the SVZ, transports Cu and other metals between the blood and the CSF [66]. The CSF is separated from the cells of the SVZ by only one layer of ependymal cells, which, to the best of our knowledge, do not form tight junctions. Thus, astrocytes have preferential (as compared with neurons) access to the interstitial fluids as well as the CSF. Observation of the extensive Cu accumulations in the astrocytes of the SVZ might lead to the idea that cells in this part of the brain function as an additional barrier between systemic fluids and brain tissues (at least in terms of Cu transport). In such a case, accumulation of Cu can be envisioned because of Cu retention at this barrier. However, we think that this is an unlikely function and that the astrocytes are instead balancing the Cu content in the brain to meet the brain’s Cu needs more than simply achieving Cu detoxification. Furthermore, the concentrations of other metals did not reach the levels revealed by copper with aging, suggesting that astrocytes are involved in regulating brain copper levels. Those findings of increased ventricular copper deposition are further extended by literature showing augmented genetic and protein expression levels of copper transporters at the choroid plexus compared to the brain parenchyma [67]. Increased copper levels have been linked to reduced neurogenesis [20], which may be another contributory factor to the vulnerability of the aged brain to neurodegenerative diseases.

Understanding the Cu binding in astrocytes is crucial for understanding the biological function and potential toxicity of increased Cu levels. Astrocytes play the important role of ‘Cu-buffering’ cells maintaining Cu homeostasis in the brain, and interruption of their functions might result in the release of toxic Cu. There is a need to conduct experiments that will explain how the Cu handling by the astrocytes is changing under various pathological conditions. Quantitative imaging techniques with subcellular resolution may be helpful for studies of the role of copper in brain functionality as well as in various pathological conditions such as brain tumors and neurodegenerative diseases.

### 2.3. Zinc

Zinc is an essential trace element with important roles in the physiology and pathology of the central nervous system (CNS). It is crucial for a wide range of cellular processes, including neurotransmission and metabolism, as it catalyses reactions of all the major classes of enzymes [68,69]. One of the primary zinc functions in the brain is its ability to modulate synaptic transmission. Zinc is found in high concentrations in synaptic vesicles of specific types of neurons, such as glutamatergic neurons in the neocortex, the amygdala and hippocampus and GABAergic neurons [70,71]. Moreover, its role in cognition and synaptic plasticity was suggested since some synaptic membrane proteins contain binding sites for extracellular Zn^2+^ [72]. Upon neuronal activity, zinc is released into the synaptic cleft, where it can modulate the activity of ion channels, neurotransmitter receptors and other signalling molecules [70]. In addition to its physiological functions, zinc has been implicated in several neurological disorders. For example, alterations in zinc homeostasis have been observed in Alzheimer’s disease, Parkinson’s disease, and other neurodegenerative disorders [73,74,75]. Zinc dysregulation has also been linked to neuronal and glial death in ischemia, epilepsy and traumatic brain injury [76,77,78].

The dynamic regulation of zinc levels in the brain is critical for maintaining proper brain function. Three families of proteins regulate its homeostasis, which include metallothioneins (MTs), zinc- and ironlike regulatory proteins (ZIPs) and zinc transporters (ZnTs) [79,80,81]. As the metallothionein have a high affinity for zinc, they are believed to be responsible for its distribution and to act as its intracellular reservoir [79,82]. Interestingly, ZnT and ZIP proteins seem to have different functions in maintaining cellular zinc homeostasis. ZnT proteins facilitate the removal of excess zinc ions from cells or transport them into intracellular vesicles, resulting in a decrease in intracellular cytoplasmic zinc levels. On the other hand, ZIP proteins promote the influx of zinc ions into cells, possibly through vesicular transport, resulting in an increase in intracellular cytoplasmic zinc levels [80].

Zinc reaches its highest concentration in the brain when compared to other organs of the human body. Interestingly, its estimated concentration is 150 μmol/L, which is 10 times higher than the concentration of zinc found in serum. Zinc plays a structural role in around 70% of proteins present in the brain, and only a small percentage (10–15%) of brain zinc exists in a “free” or cheatable form [83]. In addition, it is highly concentrated in the synaptic vesicles of a subset of glutamatergic neurons, which are also called “zinc-containing” [84]. 

The study by Ashraf et al. (2019) used synchrotron-based X-ray fluorescence microscopy to investigate the regional distribution of zinc in glia in a normal aging mouse model. It was found that zinc was concentrated in specific regions of the brain, with the highest concentration in the hippocampus, followed by the cortex and striatum, with the lowest concentration in the cerebellum. In addition, it was proven that the density of glia in the hippocampus and cortex was positively correlated with the concentration of zinc in glia. This observation suggested that glia may play a role in regulating zinc distribution in mentioned regions [17]. Previous studies have also shown that glial cells may be involved in regulating zinc levels in the brain and thus maintain the normal synapse function [21]. Moreover, MT-1 and MT-2, which act as intracellular zinc reservoirs, are primary expressed in glial cells [85,86]. 

Several studies showed that astrocytes can accumulate zinc and other toxic metals [87,88,89]. Bishop et al. (2010) has proven that rat astrocyte cultures were able to express mRNA of ZIP14, which was indicated to act as a zinc transporter across the plasma membrane of astrocytes. Thus, ZIP14 may contribute to the accumulation of zinc in these cells [90]. 

As mentioned before, ZnTs such as ZnT 1-4, play a crucial role in regulating the intracellular levels of free or loosely bound zinc [80,81,91]. Moreover, ZnT-1 which is located on the plasma membrane of neurons and glial cells, was shown to be capable of protecting the neuronal cell lines from zinc-induced toxicity [92,93,94]. Interestingly, astrocytes have been proven to have high levels of zinc-binding proteins, suggesting that these cells may be involved in regulating zinc levels in the brain [17]. Similar results were presented in the Nolte et al. study, which indicated the increased expression of ZnT-1 levels in astrocytes. This mechanism provided protection against zinc toxicity and slowed down the accumulation of intracellular zinc [18].

Several studies also confirmed the role of microglia in immune surveillance and host defence. Interestingly, neuronal injury or exposure to pathogen-derived molecules evoked microglia activation, which was associated with augmented cell proliferation and the secretion of proinflammatory mediators [95,96,97]. Extracellular zinc was indicated as one of the factors that trigger microglia activation [98]. In addition, Higashi et al. demonstrated that mouse primary cultured microglia were able to take up zinc due to metal cation transporters such as ZIP1 [99]. In CNS there is another major regulator of zinc uptake that belongs to the ZIP superfamily, called ZIP3. However, due to ZIP1’s higher abundance in brain, it is currently considered the key facilitator of neuronal zinc uptake [100,101]. Moreover, microglia may be responsible for significant physiological functions such as synaptic plasticity and neurogenesis [102]. This impact is likely due to their ability to influence MT expression and, therefore, Zn concentration [103]. 

### 2.4. Cadmium 

Cadmium belongs to the group of heavy metals with a high potential for toxicity, which is related to its half-life. Most studies put it around 15–20 years in the human body [104,105]. It was also included in the Substance Priority List 2022 (SPL), which identified the substances that could cause the most severe harm to human health due to their confirmed or suspected toxicity and the likelihood of human exposure [106]. Sources of cadmium in the human body include factories that manufacture batteries or paints, and inhalation of polluted air, cigarette smoke, or ingestion of contaminated food or water [107,108,109]. Cd is also known to be carcinogenic, and it is classified as a Group I carcinogen by the International Agency for Research on Cancer (IARC). Cadmium disrupts the DNA repair system and stimulates proto-oncogenes while inhibiting tumour-suppressor genes [108]. 

Prolonged exposure of the body to even small doses of cadmium lead to multiorgan toxicity. The human nervous system, both peripheral and central, is the most sensitive to its effects [110]. The impact of cadmium on the clinical condition of the body is broad spectrum. It includes general neurological disorders, peripheral neuropathy, olfactory dysfunction, mental retardation, learning disabilities and even behavioral changes [111]. Cadmium is also considered a factor associated with the occurrence of neurodegenerative diseases such as Alzheimer’s disease, Parkinson’s disease, and amyotrophic lateral sclerosis and as a substance playing an important role in the development of peripheral polyneuropathy [110,112]. 

The main pathological conditions in the tissues because of the action of cadmium are primarily cell apoptosis, caused by the production of reactive oxygen species, Ca^2+^ accumulation, upregulation of caspase-3 and downregulation of bcl-2, as well as p-53 deficiency [108,113]. Oxidative stress causes disruption of the microvessels in the choroid plexus in the brain tissue by wearing down the defense system against antioxidants, resulting in an increase in lipid peroxidation (LPO) and induction of the metallothionein (MT) defense mechanism. These reactions caused by cadmium may also result in the production of reactive oxygen species by mitochondria, which through the MAPKs (mitogen-activated proteinkinases) and mTOR (mammalian target of rapamycin) pathways, cause apoptosis [105,114]. Cadmium also acts on Ca^2+^ mitochondria, causing apoptosis of cortico–cerebral neurons, which has been shown in studies conducted on rats and mice. There is also a relationship between the concentration and duration of cadmium exposure and the induction of apoptosis [105,115]. 

## 3. The Influence of Metals (Fe, Cu, Zn and Cd) on Neurodegenerative Diseases

Neurodegenerative disorders are characterized by the progressive loss of selectively sensitive neuronal populations (nerve cell loss) [116]. Successive, slow and progressive dysfunction and loss of neurons and axons in the central nervous system are the main pathological features of neurodegenerative conditions—both acute and chronic—such as Parkinson’s disease and Alzheimer’s disease [117]. Neurodegenerative disease can be divided according to basic clinical symptoms (e.g., dementia), anatomical changes—the distribution of neurodegeneration (e.g., frontotemporal degeneration) or molecular abnormalities (e.g., protein abnormalities—amyloidosis, tauopathies). 

Nowadays, there is growing interest in the involvement of metal ions in neurodegenerative processes, mainly in synaptic transmission. Neurodegenerative diseases (Figure 1) e.g., Alzheimer’s and Parkinson’s, are characterized by elevated iron content in tissues and abnormal distribution of copper and zinc accumulation in amyloid [22]. 

### 3.1. Alzheimer’s Disease

Alzheimer’s disease (AD) is a neurodegenerative disorder that leads to cognitive impairment and dementia in the elderly. The main cause of the disease is mutations in the genes associated with β-amyloid (Aβ), psen1 and psen2. Many publications also emphasize the influence of various changes in lipid metabolism, endocytosis and inflammatory responses (genes: apoe, trem2, and CD33), which are inductive factors in the disease, as well as abnormal dietary habits and environmental hazards [119,120]. Changes in iron levels in AD can be diagnosed as early as mild cognitive impairment [121] using, among other things, iron contrast MRI [122,123,124,125]. Scientific evidence suggests that reduced levels of ferritin in the cerebrospinal fluid cause brain hypometabolism in people with AD [126]. Studies confirm that changes in iron metabolism trigger the disease in prodromal stages, which can be used in AD diagnosis (AD biomarker), mainly analyzing microglia. Studies confirm the apparent accumulation of iron in the plaques of activated microglia, and in some cases, even in the middle cortical layers along myelin fibers. The study shows that the degree of altered iron accumulation is clearly correlated with the amount of amyloid β plaques and tau protein pathology in the frontal cortex [127]. Interesting changes have also been observed in the brain’s immune system. Brain microglia dystrophy in the elderly and AD patients is influenced by ferritin immunoreactivity [128], which results in impaired iron homeostasis. The cause of dyshomeostasis in brain function may be, for example, brain hemorrhage due to trauma, which exacerbates astrogliosis and increases proinflammatory cytokines tumor necrosis factor- α (TNF-α) and interleukin-1β [129]. This is of particular interest because traumatic brain injury is a potential risk factor for AD [120]. 

Like iron, copper is a desirable chemical element essential for cell function, mainly due to its ability to exchange electrons. During Cu distribution in the body, copper ions undergo different states: oxidation (Cu^2+^) and reduction (Cu^+^). They bind to copper enzymes, which control a wide range of biochemical functions occurring in the body [130]. Cu is a metal ion with redox activity that is essential for aerobic organisms. As a catalytic and structural cofactor, it participates in energy production, oxygen and iron transport, cellular metabolism, peptide hormone maturation, blood coagulation, signal transduction and many other processes. The inability to control Cu balance is associated with genetic diseases (Menkes and Wilson) [131] and neurodegenerative disorders (amyotrophic lateral sclerosis, Parkinson’s disease, Huntington’s disease, encephalopathy) for which prions are responsible [132,133]. Individuals with a diagnosis of AD have been shown to have higher levels of labile serum copper than healthy individuals, with these levels correlating with poor cognitive abilities [134,135], as confirmed by postmortem analyses [136]. It is noteworthy that despite a lower total copper content, the brains of people with AD have a higher proportion of redox-active exchangeable copper, which positively correlates with increased oxidative damage and AD neuropathology [137]. With respect to neuroinflammation, copper appears to play an important role in modulating microglia activation, although there is limited evidence that it directly initiates the inflammatory process. Copper has been shown to enhance the effects of the Aβ on microglia activation and affect neurotoxicity [138]. 

Seventy percent of the proteins present in the brain contain Zn. As a structural and catalytic component, it contributes to transcription and enzymatic factors [139]. Zn homeostasis in the brain is regulated by three families of proteins: metallothioneis, which are involved in maintaining intracellular homeostasis; zinc- and ironlike regulatory proteins, which are responsible for Zn uptake from extracellular fluids into neurons and glial cells; and Zn transporters, which are associated with cellular Zn efflux [81]. In the brain, Zn is present in its free ionic form in synaptic vesicles at glutamatergic nerve terminals, from where it is released during neuronal activity [140,141]. Zn affects the expression and activity of N-methyl-3-hydroxy-5-methyl-4-isoxazolopropionate (AMPA) glutamatergic receptors, glycine ionotropic receptors and γ-aminobutyric acid (GABA) receptors [142]. Zn is intimately involved in the balance of excitatory and inhibitory signaling in the brain and is essential for memory and behavioral functions [143]. Zn homeostasis is impaired in a wide range of neurological diseases [144,145]. Although zinc does not have redox properties, excess zinc in extracellular fluid has been shown to have neurotoxic effects and affect Aβ protein aggregation [71,146]. 

Cadmium (Cd) is a carcinogenic heavy metal found in the environment. Unlike many other heavy metals, it is soluble in water and can therefore be transported from the soil to plants [147]. Once it enters the body, it accumulates in the kidneys and lower and has a half-life of 20–40 years [148,149]. Chronic exposure to Cd results in several disorders, such as hypertension, kidney dysfunction, bone demineralization and neurological diseases [150,151]. Scientific studies confirm that Cd penetrates the blood–brain barrier and accumulates in the brain, causing disorders with neurotoxicity [152]. In the brain, Cd induces activation of various signaling pathways associated with inflammation, oxidative stress, and neuronal apoptosis [153,154], leading to aggregation of Aβ plaques [155,156], blockade of M1 muscarinic receptors and phosphorylation of tau protein [157,158]. Hyperphosphorylation of tau protein destabilizes and disintegrates microtubules, disrupts axonal transport, and makes tau protein much more susceptible to aggregation in NTFs. NFT burden causes cognitive impairment and neurodegeneration, leading to the suggestion that reducing tau protein hyperphosphorylation is the key to preventing AD [159,160,161]. 

### 3.2. Multiple Sclerosis

Multiple sclerosis (MS) is an inflammatory disease of the central nervous system that causes different symptoms depending on the location: motor, sensory, visual and autonomic. The etiology of the disease is ambiguous. It is believed that it is based on genetic susceptibility and environmental factors (infection—Epstein–Barr virus, herpes virus, chickenpox virus—occupational hazards, vitamin D deficiency, obesity, smoking). Due to the different duration of disease episodes, diagnosis and leading symptoms of the disease, we distinguish different subtypes of MS: clinically isolated, relapsing–remitting, secondary progressive and primary progressive. The diagnostic parameter of the disease diagnosis is the presence of mismatched oligoclonal IgG immunoglobulins in the cerebral fluid (presence in 90% of patients) [162,163,164], which is the result of migration through the blood–brain barrier (BBB) of autoreactive T and B lymphocytes. This migration triggers several processes, including the inflammatory cascade leading to changes in microglial activation, oxidative damage and mitochondrial damage [165]. The result of these transformations are negative modifications in the myelin sheaths. A few changes are also conditioned by factors related to fluid dynamics and molecular interactions between leukocytes and vascular endothelium. Moreover, the BBB is often described as a monolithic unit, and more and more publications emphasize its heterogeneity, both in vascular and anatomical terms (location of blood vessels) [166]. Therefore, it seems highly probable that this individual vascular condition is an important element in the neuropathogenesis of MS (possibility of penetration of harmful substances causing inflammation and inducing the disease). 

Metals are indispensable cofactors of enzymes and structural elements for stabilizing static biomolecules. They participate in major metabolic pathways of the brain i.e., neurotransmitter synthesis, neural metabolism and oxygen transport [167]. In recent years, scientific attention has focused on redox active metals (Fe, Cu) due to their undeniable ability to neurodegenerate [168]. Iron and copper are of interest because of their involvement in the production of toxic reactive oxygen species (ROS). This occurs via the Fenton reaction. During this reaction, reduced iron and copper are involved in the production of hydroxyl radicals, which damage DNA, but also protein and lipids through oxidative modifications [169]. In addition, dysregulation in homeostasis of above metals contributes to several neurodegenerative diseases, including MS [170]. Numerous studies have shown that metal ions such as zinc, iron and copper enhance the aggregation of Aβ-amyloid, α-synuclein, prion protein and ataxin-3, thereby inducing neurodegeneration [171]. 

### 3.3. Parkinson’s Disease

Parkinson’s disease (PD) is a self-limiting, progressive degenerative disease of the central nervous system (CNS) [172]. During PD, degenerative changes occur within nerve cells in the substantia nigra and other pigment-bearing aeras of the brain. Substantia nigra neurons are responsible for the production of dopamine, a neurotransmitter, hence they are also called dopaminergic neurons [173]. The dysfunction of dopaminergic neurons is associated with a deficiency of dopamine in the substantia nigra and striatum, resulting in a preponderance of activity of glutamatergic neurons, which inhibit the thalamic nuclei [174]. The association with aging of dopaminergic neurons leads to motor complications such as tremor, rigidity, motor slowing and postural abnormalities [175].

As mentioned earlier, in Parkinson’s disease there is a loss of dopaminergic neurons in the substantia nigra pars compact (SNpc). During PD, zinc and iron levels are increased, while copper levels in the substantia nigra are reduced [22]. An excess of the above elements destroys neurons, because of which the function of lysosomes i.e., cellular structures responsible for digesting and repairing damaged proteins, is impaired. It is normal that as we age, the function of lysosomes is weakened, which translates into a slowing down of their work and thus the process of renewal of organic material is impaired. Damaged, undigested protein can consequently overbuild in cells and allow iron to reach nerve cells and cause oxidative stress [176]. 

Previous studies have shown neurotoxic effects of copper, iron and manganese, while zinc can have a bidirectional effect, i.e., neurotoxic but also neuroprotective depending on the dose and disease state. It was suggested that increased copper levels can contribute to oxidative stress and neuroinflammation, which are implicated in neurodegeneration. It was due to the activation of redox-sensitive transcription factors. In addition, reduced ceruloplasmin levels may lead to increased accumulation of iron in the brain. As a result, elevated iron levels and accumulation of redox-active Fe^2+^ may cause metal-induced oxidative stress [177]. Thus, conclusions have been drawn suggesting that metals are involved in neurodegeneration through modulation of protein aggregation. Exposure to heavy metals significantly increases α-synuclein synthesis and aggregate formation, which is through to be one of many etiopathological factors in PD. Another report states that a major factor in disease progression (PD) is inflammation correlated with dysfunction of metal iron (Fe) homeostasis accompanied by oxidative stress [178]. In addition, it was also shown that the Fe accumulation reduced the activity of Fe-dependent hypoxia-inducible factor (HIF-1α), resulting in decreased levels of tyrosine hydroxylase (TH) and dopamine. The decline in these biomarkers ultimately leads to the degeneration of dopaminergic neurons [179]. The role of Fe as a risk factor in PD progression was suggested when total Fe levels increased by 176% and iron ion levels by 225% in SNpc compared to age-matched controls. Animal models of PD have demonstrated neuroprotection by genetic or pharmacological chelation of Fe [180]. Spectrophotometric and Pearl staining inductively coupled plasma spectroscopy, MRI, laser microprobe mass analysis, sensitivity-weighted imaging (SWI) and enhanced T2-star angiography (ESWAN) showed elevated Fe levels in brain SNpc [181]. 

As mentioned before, previous biochemical investigations of brain samples from individuals with PD have shown reduced levels of Cu in the substantia nigra and promotion of α-synuclein aggregation. The expression of Cu transporter 1 (Ctr1), responsible for Cu uptake, was also significantly decreased in PD, suggesting its association with the disease [182]. Interestingly, Cruces-Sande et al. showed that the nigrostriatal administration of copper alone did not induce degeneration of dopaminergic neurons. However, when combined with 6-hydroxydopamine (6-OHDA) in a rat model of Parkinson’s disease (PD), copper potentiated the neurodegenerative effects caused by 6-OHDA. This enhanced degeneration was also associated with increased oxidative stress [183].

The exposure to Cd also resulted in oxidative stress. The increase in ROS and free radicals was shown to result in aggregation of α-synuclein and the release of proinflammatory molecules including IL-6 and TNF-α. However, the levels of IL-10 were decreased. As a result of neuroinflammation, the activity of the CREB pathway was inhibited which was associated with neurodegeneration and neurogenesis decline [184]. In addition, persistent activation of signaling pathways such as MAPK and mTOR is associated with cell death and neuronal apoptosis in response to Cd exposure [185]. Cd influence on neuronal death causing the neurodegenerative diseases was particularly observed in cortical and hippocampal regions [186].

In addition, Arsenic (As) is also believed to contribute to the development and progression of PD through various mechanisms. This includes oxidative stress induction, which is characterized by elevated levels of reactive oxygen species (ROS) and lipid peroxides [187]. The neurotoxic effects of As involve the impairment and degeneration of dopaminergic neurons due to oxidative DNA impairment [188]. Exposure to As leads to the release of αSyn, potentially connecting with disruptions in the ubiquitin–proteasome system, oxidative damage or impaired mitochondrial function, ultimately promoting the degeneration of neurons and triggering cell death [189].

Considering the above issues, oxidative stress plays an important role during PD—both chronic and spontaneous. It is suspected that free radicals are associated with dopamine, and this is since enzymes, such as monoamine oxidase, are involved in the dopamine metabolic pathway. Hence, dopamine and its metabolites are involved in the production of free radicals [190]. According to Hassanzadeh et al. (2018), there are several phenomena observed during Parkinson’s disease, i.e., an increase in iron levels, impaired function of the mitochondrial complex I, a decrease in glutathione levels and a decrease in 26S proteosome activity [191]. 

Experimental studies have proven that heavy metal exposure at a young age may have long-term neurological and epigenetic effects, suggesting that heavy metal-exposed children are at higher risk of developing neurodegenerative diseases in the future. It was shown that metals play an important role in regulating epigenetics during PD [192,193,194], but studies have revealed that the environmental factors play a greater role than genetic factors [195,196]. Metal-induced neurotoxicity in PD is still under research. A summary of the effects of selected metals during Parkinson’s disease is shown below in Table 1. 

### 3.4. Amyotrophic Lateral Sclerosis (ALS)

ALS is a neurodegenerative, progressive disease, which affects mainly motor neurons [204]. The typical disease course begins with weakness of the distal limbs’ muscles, which eventually leads to paralysis, but there can also occur a bulbar type of ALS, where the dominating symptoms are dysarthria and dysphagia [205]. ALS is one of the most rapidly progressive neurodegenerative diseases, as the patients on average live up to 3 years after the first symptoms occur [205]. Most of the cases are sporadic, but there is also a familial type of ALS, which is associated with genes’ mutations [206]. One of the changes that could appear in the familial ALS type is the *SOD1* gene mutation. It encodes an enzyme called Cu/Zn superoxide dismutase, which acts as an antioxidant and requires copper and zinc to work properly [207]. There is still much to learn about this gene and its mutation, but some research conducted on animal models gives hope that this area might be helpful in ALS treatment [207].

Besides familial ALS, the etiopathogenesis of the sporadic types of the disease remains unknown. Currently, there are no discovered biomarkers that could indicate the possibility of the disease occurring within people [208]. The possible connection between trace elements abnormalities and ALS is under consideration. For example, some research showed increased iron accumulation in the neurons of the ALS-affected patients, especially in microglia, which, due to the Fenton reaction, may induce an oxidative stress in the cells, causing damage to these neurons [17,27,28,209]. Oxidative stress can also be induced by high cadmium [29] and copper levels [12,17], or by low and high zinc levels [12,17].

Currently, clinical trials are being held to develop a new, successful ALS treatment, involving pharmaceutical and gene therapy [210]. Some therapies consider trace-elements disorders and their possible outcomes as an anchor point. For example, edaravone treatment aims to reduce the oxidative stress in the neurons, as well as regulating-activated microglia [211]. There have been attempts to use chelators as a therapy, but this kind of treatment still needs more research to show its usefulness in ALS patients [209]. Deferiprone is an iron chelator, which is currently being tested as a possible neuroprotective drug in ALS [210,212].

Both etiopathogenesis and the treatment of ALS remain an aim for many researchers. Many facts are yet to be discovered. There is a space for new clinical trials, which would help in developing a successful treatment for this rapidly progressive neurodegenerative disease.

## 4. Discussion

In recent years, prominent improvements have been made in understanding metal metabolism in glial cells. In this review, we have summarized the data on the role of iron, copper, zinc and cadmium metabolism and distribution in glia and discussed their possible involvement in some neurodegenerative diseases. Metal dyshomeostasis in glial cells are pervasive features of normal aging, rendering the brain responsive to neurodegenerative diseases. Most noticeably, differential distributions of the metals in glia of the various brain regions were apparent. Meanwhile, copper seemed to show modest increments in their concentration confined to the astrocytes of globus pallidus and zinc was enriched in the glia of hippocampus and cortex. Taken together, we summarize that changes in glial dystrophy with aging may induce differential regional content of various metals and are a sign of the aging brain, which can also take part in neurodegenerative diseases.

Previous studies have shown that higher concentrations of iron are present in the substantia nigra and globus pallidus [213]. Furthermore, the levels of iron increase with age, particularly in the basal ganglia, indicating that iron is an age-dependent metal in the brain [37]. The elevated iron levels can be attributed to continuous iron delivery to the brain and reduced iron release into the bloodstream through the blood–brain barrier (BBB) and blood–cerebrospinal fluid barrier (BCSFB) [16]. Oligodendrocytes, a type of brain cell, contain the highest amount of iron [44]. Iron plays a crucial role in fueling microglia, another type of brain cell, to prevent inflammation, but excessive iron accumulation in microglia can accelerate the aging process and increase susceptibility to neurodegenerative diseases. Interestingly, Ashraf et al. (2019) [17] discovered that the ratio of iron to microglia increased in the basal ganglia with age, while the iron to astroglia ratio was elevated in the substantia nigra and globus pallidus but decreased in the striatum. These findings suggest that glial cells may experience abnormal signaling, leading to age-related metal imbalances in metal levels and contributing to neurodegenerative processes. Studies have confirmed that changes in iron metabolism can trigger disease progression in prodromal stages and can be utilized as a biomarker for Alzheimer’s disease (AD), particularly by analyzing microglia. Notably, iron accumulation has been observed in activated microglia within plaques, and in some cases, even in the middle cortical layers along myelin fibers. 

According to research, ref. [17] copper levels were higher in the striatum, cingulate cortex and ventral hippocampus compared to the globus pallidus in 6-month-old mice. Moreover, the level of copper was higher in the ventricles with aging [22,63]. It was insinuated that the BCSFB is the predominant barrier for copper uptake in the brain [64]. Astrocytes that are close to ventricles are in a prime location to balance brain copper content and achieve detoxification, having access to CSF [20]. Therefore, the concentrations of other metals did not reach as high levels demonstrated by copper with aging, suggesting that astrocytes are mostly involved in regulating brain copper levels. As increased copper levels have been linked to reduced neurogenesis [20] which may be another conducive factor to the sensitivity of the aged brain to neurodegenerative diseases. Individuals with a diagnosis of AD have been shown to have higher levels of labile serum copper than healthy individuals, with these levels correlating with poor cognitive abilities [134,135], as confirmed by postmortem analyses [136].

Studies have revealed that zinc accumulation is higher in the hippocampus compared to other brain regions [78,143], indicating its significance in learning and memory consolidation. At two months of age, zinc levels were found to be higher in the globus pallidus than in the substantia nigra, although comparable to those in the hippocampus [17]. Zinc has been shown to modulate GABAergic transmission in the globus pallidus [143] and is believed to play an important role during neurodevelopment in this brain region. Brain zinc levels have been reported to increase with aging in both rats [214] and humans [215]. Interestingly, zinc accumulation has been observed in the globus pallidus in a rat model of Parkinson’s disease (PD) induced b 6-hydroxydopamine [54], as well as in the substantia nigra and stratium of individuals with PD [168]. Thus, the higher zinc level in the globus pallidus of aged mice suggests that zinc dysregulation may contribute to aging as a major risk factor for neurodegenerative diseases. Glial cells, specifically astrocytes, play a crucial role in maintaining zinc homeostasis to ensure optimal synaptic signaling [21]. Ashraf et al. (2019) [17] found an association between zinc and glial cells in the globus pallidus, demonstrating that astrocytes accumulate zinc [18]. Microglia can uptake zinc through the zinc transporter ZIP1, which triggers sequential microglial activation [99]. The ratio of zinc to astroglia was reduced in the basal ganglia during older ages. In Parkinson’s disease, zinc and iron levels are increased, while copper levels in the substantia nigra are reduced [22]. Previous studies have indicated neurotoxic effects of copper, iron, and manganese, while the effects of zinc can be bidirectional, with both neurotoxic and neuroprotective properties, depending on the dosage and disease state [177].

Due to cadmium neurotoxicity, even small doses of this metal could damage the brain parenchyma. Cadmium is known to accumulate in the brain of young and adult animals [152]. In addition, Cd also increases the exposure of the brain to other potentially dangerous substances [152]. Several published in vivo studies have shown a range of possible effects of Cd on glial cells, depending on dosage, time of exposure and experimental model used [216,217]. In mammals, cadmium impairs GFAP expression in the astroglia. The process of reactive astrogliosis is well known and is characterized by overexpression of GFAP, hypertropia and proliferation of astroglia with subsequent formation of glial scar [218]. These studies were also reported in the lizard and zebrafish brains, in which histological alterations, reduction of GFAP expression and bioaccumulation were observed after the exposition on Cd [217,219]. Moreover, this metal has chemical similarity to zinc and calcium, and it can disrupt the metabolism of those elements [220]. 

However, the current research on metal metabolism relies mainly on experimental animals. These models cannot fully imitate changes of iron, copper, zinc, cadmium metabolism in the brain of patients with neurodegenerative diseases. Therefore, it is crucial for studies using human glia, human stem cells or postmortem brain tissue of patients with Alzheimer disease, multiple sclerosis, Parkinson disease and Amyotrophic lateral sclerosis to clarify metal metabolism of glia and their role in neurodegenerative diseases.

The nervous system does not regenerate as well as other systems do. Neurodegeneration usually become progressive with age, as typically seen in AD and PD. With the increase of lifespan among the general population, there is a longer duration of exposure to metals for individuals and potential increase in incidence of neurological diseases. Future experimental studies need to focus on the synergic effect of metal mixture exposition. Humans are exposed to multiple metals at the same time and neurotoxicity is commonly accompanied with metal over exposition. Further studies are needed to explore the health impact of metal mixture for better understanding of their synergetic and antagonistic effect. It is also important to identify metal-specific transporters as well as design therapeutics to treat metal-induced toxicity.

## Figures and Tables

**Figure 1 brainsci-13-00911-f001:**
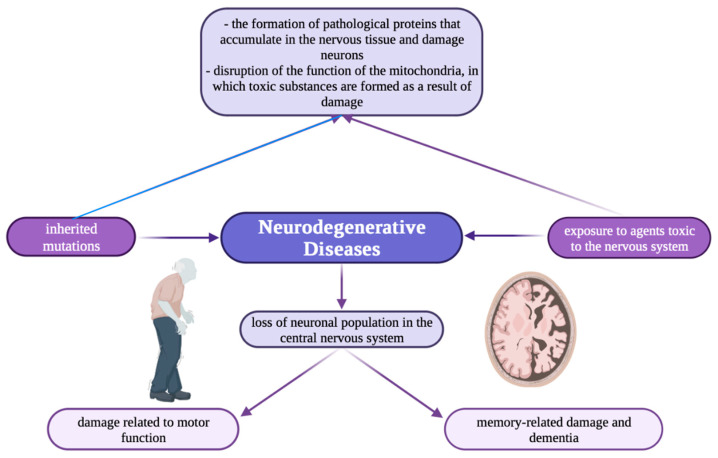
Schematic illustration showing the mechanism of neurodegenerative diseases [118].

**Table 1 brainsci-13-00911-t001:** Effects on Parkinson’s disease of selected heavy metals.

Type of Element	Impact on Parkinson’s Disease	Ref.
Iron (Fe)	During Parkinson’s disease, iron metabolism in the extrapyramidal system is abnormal. The increase in iron levels may be related to age or disease-causing loss of proteins that could store iron. Another possibility could be increased import (increased expression of transferrin receptor 1 and divalent metal transporter 1) or decreased export (decreased expression of ferroportin-1) during Parkinson’s disease and activation of microglia in response to neurodegeneration. Available research results indicate that every patient with PD has a disruption in homeostasis between the iron in the blood and in the brain.	[197,198]
Copper (Cu)	The epidemiological correlation between chronic copper exposure and a higher risk of developing PD is well known. However, copper dyshomeostasis in PD, as a cofactor in the active sites of several enzymes and thus a participant in many enzymatic intracellular reactions, is relatively new. Cu is a cofactor of superoxide dismutase 1 (SOD-1), an essential component of ceruloplasmin, a ferroxidase that oxidizes reactive iron Fe^2+^ to the nontoxic Fr^3+^ form. Reduced Cu levels can thus lead to less efficient removal and increased production of reactive oxygen species, resulting in increased oxidative stress during PD. In addition, Cu is a component of cytochrome c oxidase which is responsible for electron transfer of the mitochondrial pathway. Therefore, reduced Cu levels may impair the function of cytochrome c oxidase, thereby impairing mitochondrial function.	[199,200,201]
Zinc (Zn)	Synaptic Zn^2+^ mediates its effects mainly by altering synaptic transmission in the striatum. Available findings suggest that synoptically released Zn^2+^ from corticospinal striatal terminals may play a deleterious role by promoting the expression of motor and cognitive deficits associated with Parkinson’s disease.	[202]
Cadmium (Cd)	Cadmium has an extremely long biological half-file. Cd-dependent neurotoxicity has been linked to Parkinson’s disease. At the cellular level, Cd affects cell differentiation, proliferation, and apoptosis.	[111,203]

## Data Availability

Not applicable.

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
