# Peer review of "Distribution of Iron, Copper, Zinc and Cadmium in Glia, Their Influence on Glial Cells and Relationship with Neurodegenerative Diseases"

_brainsci, 2023, doi:10.3390/brainsci13060911_

Round 1
Reviewer 1 Report
Comments and Suggestions for Authors
The content is interesting but It would be complementary to put a figure of the effect of metals on glia cells
Author Response
Dear Reviewer,
Thank you very much for reviewing our manuscript. We appreciate the interest and commitment you have provided for this work. We are very grateful for your extremely precious comments. We are convinced that thanks to your suggestions this manuscript will be much more valuable.
We are pleased to submit explanations and details of our revisions in the manuscript entitled “Distribution of iron, copper, zinc, cadmium in glia, their influence on glial cells and relationship with neurodegenerative diseases”.

Reviewer 2 Report
Comments and Suggestions for Authors
Review of a manuscript “Distribution of iron, copper, zinc, cadmium in glia, their influence on glial cells and relationship with neurodegenerative diseases” by Aleksandra Górska and coauthors submitted to “Brain Sciences”.
In the manuscript the authors review the data on the distribution and influence of iron, copper, zinc, and cadmium in glial cells and their association with neurodegenerative diseases. The review also considers some aspects of the role of metals in physiology and pathophysiology of the human brain. This is an important area of biomedical research, and the discussions and hypothesis of this manuscript will be interesting for the readers of the journal.
The following corrections and additions should be made.
Abstract
Line 18 “In the article, the distribution and influence of copper, zinc and cadmium in glial cells were investigated.” Since this is a review manuscript, it will be more appropriate to write: ”Recent data on the distribution and influence of copper, zinc and cadmium in glial cells are summarized”.
Line 22 “We have collected the data from various sources to present the latest results in the article”. This sentence contains an obvious statement and can be deleted.
Line 26:” Thus, conclusions have been drawn suggesting that metals are involved in neurodegeneration through modulation of protein aggregation.” The sentence beginning with “Thus” sounds as a summary of previous statements, however there is nothing about protein aggregation. It should be corrected, for example as follows:” Recent data points to the association of metals with neurodegeneration through their role in the modulation of protein aggregation.”
Introduction
Line 36: “…as they play part in some important mechanisms [1-7]” should be corrected as follows :”..as they play a role in some important mechanisms [1-7]”
3. The influence of metals (Fe, Cu, Zn, Cd) on neurodegenerative diseases
Lines 338-339 “Ryc.1 Schematic illustration showing the mechanism of neurodegenerative diseases [119]. What is “Ryc 1” ? It is also unclear what images are located on the left and right of the “loss of neuronal population in the central nervous system”. It should be explained in the figure legend.
3.1 Alzheimer’s disease
Lines 341-372 contain well know general information about the pathology of Alzheimer’s disease. This should be truncated, since it contains common knowledge not directly associated with the role of metals.
3.3 Parkinson’s disease
Line 482. “Parkinson’s disease (PD) is a self-limiting, progressive degenerative disease of the central nervous system (CNS). “ The authors should add a relevant reference on a recent review on PD : ”Biomarkers in Parkinson’s Disease”. Chapter in a book, Peplow PV et al., (eds) Neurodegenerative Diseases Biomarkers. 2022. Neuromethods, vol 173. pp 155-180. Humana, New York, NY. https://link.springer.com/protocol/10.1007/978-1-0716-1712-0_7
Discussion
Lines 559-560: ”In this review we have summarized iron, copper, zinc and cadmium metabolism and distribution in glia and reviewed their possible roles in the selected neurodegenerative diseases.” The sentence should be corrected as follows: ”In this review we have summarized the data on the role of iron, copper, zinc and cadmium metabolism and distribution in glia and discussed their possible involvement in some neurodegenerative diseases.”
Lines 631-632:” Recently, it was discovered that the radial glia cells at ventricular level are involved in the processes of adult neurogenesis and regeneration of nerve tissue in many species of vertebrates [221]. The authors should explain how this finding is related to metals.
Author Response

(The authors gave the same response as above.)

Reviewer 3 Report
Comments and Suggestions for Authors
The manuscript is well written and introduces the topic of new research but lacks in several areas; these issues must be addressed to make it interesting for the journal readers.
Point 1: Abstract: There is a necessity to draw the graphical abstract to show an increase or decrease in heavy metals involved in regulating neuronal functions that impact the neurodegeneration process.
Point 2: Introduction: There is a need to rewrite the introduction. It was not focusing on the glial cells.
Point 3: There is a need to establish the relationship between heavy metals such as copper (Cu), arsenic (As), cadmium (Cd), iron (Fe), and lithium (Li) in neurodegeneration and how it impacts the pathological conditions of PD.
Point 4: This review did not discuss the role of heavy metals on DA receptors. Why?
Point 5: There is a lack of summarizing the influence of heavy metals on PD in Tabular form.
Point 6: There are many reviews available on this topic. What is the novelty of this review?
Point 7: Discuss the Effect of heavy metals on genetic alterations in PD and neurodegeneration.
Point 8: Add the conclusion with prospects of research in this field.
Author Response

(The authors gave the same response as above.)

Round 2
Reviewer 3 Report
Comments and Suggestions for Authors
· Most of the suggestions have been incorporated by the authors in the revised manuscript. Therefore, no issue with considering it for publication.